# Synergistic Effects of Lenvatinib (E7080) and MEK Inhibitors against Anaplastic Thyroid Cancer in Preclinical Models

**DOI:** 10.3390/cancers13040862

**Published:** 2021-02-18

**Authors:** Keisuke Enomoto, Shun Hirayama, Naoko Kumashiro, Xuefeng Jing, Takahito Kimura, Shunji Tamagawa, Ibu Matsuzaki, Shin-Ichi Murata, Muneki Hotomi

**Affiliations:** 1Department of Otolaryngology—Head and Neck Surgery, Wakayama Medical University School of Medicine, 811-1 Kimiidera, Wakayama 641-8509, Japan; kenomoto@wakayama-med.ac.jp (K.E.); shunhira@wakayama-med.ac.jp (S.H.); nk9046@wakayama-med.ac.jp (N.K.); xfjing@wakayama-med.ac.jp (X.J.); abo1201@wakayama-med.ac.jp (T.K.); tamashun@wakayama-med.ac.jp (S.T.); 2Department of Human Pathology, Wakayama Medical University School of Medicine, 811-1 Kimiidera, Wakayama 641-8509, Japan; m_ibu@wakayama-med.ac.jp (I.M.); smurata@wakayama-med.ac.jp (S.-I.M.)

**Keywords:** lenvatinib, E7080, selumetinib, AZD6244, combination therapy, thyroid cancer

## Abstract

**Simple Summary:**

Lenvatinib has been found to be effective against radioiodine-refractory thyroid cancer. However, compliance with lenvatinib therapy is poor due to cancer progression and adverse effects. To improve the success rate of lenvatinib treatment, we propose a combination of lenvatinib and another drug class that has a different target than lenvatinib. We tested the potential of this combination in preclinical models and found that it is effective in our models. The combination also does not cause significant adverse effects in our mouse model. Thus, our results suggest that the combination we propose in this study is a potential therapeutic approach for thyroid cancer that is not responsive to radiotherapy.

**Abstract:**

E7080, known as lenvatinib, is an oral multitargeted tyrosine kinase inhibitor that has been shown to improve the survival rate of patients with radioiodine-refractory thyroid cancer. However, a majority of patients do not continue lenvatinib intake due to disease progression or significant toxicity. To improve treatment success rates, we propose the combination of lenvatinib with mitogen-activated protein kinase (MEK) inhibitors. To test this hypothesis, we tested the effects of lenvatinib with the MEK inhibitor U0126 in vitro using two human anaplastic thyroid cancer (ATC) cell lines, 8505C and TCO1, and with another MEK inhibitor, selumetinib (AZD6244), in an ATC mouse model. We found that the combination of lenvatinib with MEK inhibitors enhanced the antitumor effects of monotherapy with either agent in vitro and in vivo, and these effects may be through the AKT (Protein Kinase B) and extracellular signal-regulated kinase (ERK) signaling pathways. Furthermore, the combination does not have significant adverse effects in the ATC mouse models in terms of body weight, blood biochemical parameters, and histopathology. In conclusion, the combination of lenvatinib with an MEK inhibitor is a potentially viable therapeutic approach for ATC treatment.

## 1. Introduction

Thyroid cancer (TC) is the most common endocrine carcinoma, with reported incidence of 12.9 per 100,000 individuals in the United States (2011–2015) and 25.8 per 100,000 individuals in Puerto Rico (2011–2015) [1]. The incidence of TC has substantially increased in several countries, and this increase is attributed to diagnostic imaging and fine-needle aspiration [2]. Most differentiated thyroid cancers (DTCs), such as papillary thyroid cancer (PTC) and follicular thyroid cancer (FTC), are characterized by slow growth and mild invasion, with an overall survival rate of over 90% within 10 years [3,4,5,6,7]. However, the therapeutic options for TC after conventional treatment, such as surgery and following radioiodine therapy, are limited; TC that persists after this line of treatment is called radioiodine (RAI)-refractory TC. Among the RAI-refractory TCs, anaplastic thyroid cancer (ATC) is the most aggressive, displaying rapid tumor growth, local invasion, and frequent distant metastasis, and the 6-month survival rates for ATC were reported to be only 26.6% and 6.6% for TNM stages IVB and IVC, respectively [8,9,10].

Molecular marker-guided TC management was developed recently [11]. To target RAI-refractory TC, a novel molecularly targeted therapy, E7080 (lenvatinib), has been approved by the US FDA and has been effective in improving progression-free survival (PFS) and response rates in patients [12]. Lenvatinib is an oral multitarget tyrosine kinase inhibitor (TKI) that prevents signaling through several molecular pathways for tumoral angiogenesis, including those that involve vascular endothelial growth factor receptor (VEGFR) 1–3, fibroblast growth factor receptor (FGFR) 1–4, platelet-derived growth factor receptor (PDGFR) β, stem cell factor receptor (KIT), and rearranged during transfection (RET) [13,14,15]. However, continuous use of lenvatinib has been associated with the appearance of toxicity, such as hypertension, diarrhea, fatigue, appetite loss, and proteinuria [16].

To gain maximum benefit for overall survival, it is essential to understand the molecular mechanism of lenvatinib’s antitumor effect for the RAI-refractory TCs such as ATC. Furthermore, the establishment of a combination therapy of lenvatinib will drastically improve ATC patients’ survival chances in case of dose reduction due to toxicity. In this study, we describe the mechanism that underlies the effects of lenvatinib against RAI-refractory TC using human ATC cell lines. Moreover, we explore the potential of combinatorial therapy for ATC treatment with lenvatinib and mitogen-activated protein kinase (MEK) inhibitors in vitro and in vivo.

## 2. Materials and Methods

### 2.1. Cell Culture

Two human ATC cell lines, 8505C and TCO1, were used in this study. The 8505C and TCO1 cells were purchased from the Japanese Collection of Research Bioresources (JCRB) Cell Bank (Osaka, Japan), and a mycoplasma test was performed by the International Council for Laboratory Animal Science Monitoring Center (Kawasaki, Japan) before the experiments were performed. These two cell lines have different genetic characteristics that have been detailed by Ito et al., Zhang et al., and Landa et al. [17,18,19]. In brief, 8505C has TP53, B-Raf proto-oncogene, serine/threonine kinase (BRAF), epithelial growth factor receptor (EGFR), and phosphoinositide-3-kinase regulatory subunit 1/2 (PI3K3R1/2) mutations, while TCO1 has TP53, BRAF, neurofibromin 2 (NF2), and telomerase reverse transcriptase (TERT) mutations. The cells were cultured in RPMI-1640 media (Thermo Fisher Scientific, cat. 61870-036) supplemented with 10% fetal bovine serum and 1% antibiotic–antimycotic solution (Thermo Fisher Scientific, cat. 15240-062) in 5% CO_2_ at 37 °C in a humidified incubator. The medium was replaced every 48 h.

### 2.2. Inhibitors

E7080 (lenvatinib) was purchased from AdooQ BioScience (Irvine, CA, USA). Selective MEK inhibitors, U0126 for the in vitro study and AZD6244 (selumetinib) for the in vivo study, were purchased from MedChemExpress (Monmouth Junction, NJ, USA) and Selleck Chemicals (Houston, TX, USA), respectively.

### 2.3. Western Blot

Western blot analyses were carried out as previously described [20]. In brief, the cultured cells were washed with phosphate-buffered saline (PBS) and homogenized in a solution containing 50 mM Tris buffer, 150 mM NaCl, 1 mM ethylenediaminetetraacetic acid, 0.1% Triton-X 100 (Sigma-Aldrich, St. Louis, MO, USA), 1% NP40, and proteinase/phosphatase inhibitors (Protease/Phosphatase Inhibitor Cocktail #5872, Cell Signaling Technology, Danvers, MA, USA). After centrifugation at 16,000 rpm for 10 min, lysates were used for Western blot analysis. Protein concentration was calculated using the Pierce Coomassie Plus Protein Assay Kit (Thermo Fisher Scientific, Cambridge, MA, USA) according to the manufacturer’s protocol and the BioPhotometer plus (Eppendorf, Hamburg, Germany). The cell lysates were subjected to sodium dodecyl sulfate–polyacrylamide gel electrophoresis (SDS-PAGE), transferred to polyvinylidene fluoride membranes, and immunostained using an iBind Western Device (Thermo Fisher Scientific) with antibodies against phospho-AKT (1:2000 dilution; #4060), AKT (1:1000 dilution; #4691), phosphorylated extracellular signal-regulated kinase (ERK; 1:2000 dilution; #4370), ERK (1:1000 dilution; #4695), α-tubulin (1:4000 dilution; #3873), and cyclin D1 (1:1000 dilution; #55506) purchased from Cell Signaling Technology. Horseradish peroxidase-conjugated anti-mouse immunoglobulin G (IgG) (1:1000 dilution; #7076) or anti-rabbit IgG (1:1000 dilution; #7074) from Cell Signaling Technology was used as the secondary antibody. Immunostaining signals were detected using an enhanced chemiluminescence (ECL) system (Amersham ECL Prime, GE Healthcare, Chicago, IL, USA). To detect α-tubulin as a loading control and to detect total AKT and ERK, the blots were stripped with Re-Blot Plus (Millipore Sigma, Temecula, CA, USA) and re-probed for total form proteins and α-tubulin. ImageJ software (version 1.52v; Wayne Rasband, National Institutes of Health, Bethesda, MD, USA) was used for quantification of band intensities. All uncropped western blot figures can be seen in Appendix A.

### 2.4. Immunofluorescence Staining

8505C and TCO1 human ATC cells were cultured in the medium with 50 μM lenvatinib or with dimethyl sulfoxide (DMSO; control). After 24 h, the cells were fixed in a culture well with 4% paraformaldehyde for 30 min. The cells were permeabilized with 100% ethanol for 10 min and 0.25% NP40 for 5 min and blocked with 10% donkey serum for 1 h. The primary antibodies used were anti-FGFR1 (1:200 dilution; #9740), anti-PDGFRβ (1:100 dilution; #3169), anti-VEGFR2 (1:800 dilution; #9698), anti-cleaved caspase 3 (1:400 dilution; #9661), and anti-α-tubulin (1:1000 dilution; #3873) from Cell Signaling Technology. After three washes with PBS, the cells were incubated for 1 h with Alexa Fluor 488-conjugated anti-mouse (1:1000 dilution; #4408) and Alexa Fluor 555-conjugated anti-rabbit (1:1000 dilution; #4413) antibodies purchased from Cell Signaling Technology. 4,6-diamidino-2-phenylindole (DAPI) at a concentration of 1 mg/mL was used for nuclear visualization and was added at the end of the process. The images were obtained through fluorescence microscopy (NIKON, Tokyo, Japan).

### 2.5. XTT Assay

The Cell Proliferation Kit II (XTT) assay (Roche, cat. 11465015001; St. Louis, MO, USA) was used according to the manufacturer’s protocols. Briefly, cells were grown in 96-well microplates in a final volume of 100 μL culture medium per well. The cells were incubated with 0.1 to 50 μM lenvatinib for 48 h. Then, 50 μL of the XTT labeling mixture was added to each well to a final XTT concentration of 0.3 mg/mL. After incubation of the microplate for 4 h in 5% CO_2_ at 37 °C in a humidified incubator, the formazan dye formed was quantified using a scanning multiwell spectrophotometer (SH-9000 microplate reader, Corona, Hitachinaka, Japan).

### 2.6. Cell Proliferation Assays

To evaluate the synergic effects of lenvatinib and U0126 in inhibiting cell proliferation, 8505C and TCO1 cells were cultured in the medium with increasing concentrations of lenvatinib plus E7080, lenvatinib (5 μM), U0126 (5 μM), or DMSO (control) in 6-well plates in triplicates. Cell counts were measured every 24 h for 96 h using a Countess automated cell counter following the manufacturer’s protocol (Invitrogen, Carlsbad, CA, USA).

### 2.7. In Vivo Xenograft Tumor Assays

All animal experiments were performed according to protocols approved by the Animal Ethics Committee of the Graduate School of Wakayama Medical University (#1012). Female athymic nude mice, aged 6 to 8 weeks, were used for the xenograft assays. 8505C cells (1 × 10^6^ cells) mixed with Matrigel basement membrane matrix (BD Biosciences, cat. 354234; San Jose, CA, USA) in 200-μL suspension were inoculated subcutaneously into the right flank of each mouse. The mice were randomly divided into four groups: control group (*n* = 6), lenvatinib (30 mg/kg/day) monotherapy group (*n* = 3), selumetinib (30 mg/kg/day) monotherapy group (*n* = 3), and lenvatinib (30 mg/kg/day) plus selumetinib (30 mg/kg/day) combination therapy group (*n* = 3). Lenvatinib and selumetinib were first dissolved in DMSO and diluted in 4 volumes of 0.5 w/v% methyl cellulose 400 solution (FUJIFILM Wako Chemicals Corporation, cat. 131-17811, Osaka, Japan) and then orally administered to the mice.

The tumor volume was calculated as L × W × H × 0.5236 mm. Treatment began when the median tumor size reached approximately 100 mm^3^. Tumor growth rate was calculated using the following equation:(1)tumor growth rate = (V2 − V1)/(t2 − t1)
where V2 is tumor volume at euthanasia, V1 is tumor volume when treatment was started, t2 is the day of euthanasia, and t1 is the day when treatment was started.

Treatment was continued until the first mouse reached the humane endpoint criteria such as maximum tumor diameter > 20 mm and unacceptable pain, upon which all mice were euthanized using isoflurane inhalation. Then, the tumor tissues and blood serum were collected for further analysis. Formalin-fixed paraffin-embedded sections from the internal organs were analyzed by hematoxylin and eosin (H&E) staining according to standard methods. All slides were reviewed by pathologists and photographed using a NIKON Microscope with standard software. Serum total protein (TP), albumin (ALB), lactate dehydrogenase (LDH), triglyceride aspartate aminotransferase (AST), alanine aminotransferase (ALT), bilirubin blood urea nitrogen (BUN), creatinine (CRE), and amylase (AMY) levels were quantified through routine laboratory methods (Nagahama Life Science Laboratory, Shiga, Japan).

### 2.8. Immunohistological Analysis

Immunohistochemistry (IHC) was performed using Discovery Auto-Stainer with automated protocols from formalin-fixed paraffin-embedded mouse sections (Ventana Medical Systems, Inc., Tucson, AZ, USA; Roche, Mannheim, Germany). The primary antibodies used were anti-Ki67 antibody (#518–102456) from Roche (Mannheim, Germany) and anti-cleaved caspase 3 antibody (1:200 dilution; #9661) from Cell Signaling Technology.

### 2.9. Statistical Analysis

Data are expressed as the mean ± standard error (SE). All tests were two-sided, and *p* < 0.05 was considered statistically significant. GraphPad Prism version 8.0 (GraphPad Software, La Jolla, CA, USA) was used to perform analyses of the variance. Kaplan–Meier curves of cause-specific survival (CSS) and adherence ratio were drawn for 25 patients from the date of initial lenvatinib administration to the date of cause-specific death or last contact.

## 3. Results

### 3.1. Lenvatinib (E7080) Suppresses AKT Signaling and Induces Apoptosis

To confirm the expression of lenvatinib-targeted molecules (FGFR1, PDGFRβ, and VEGFR2) in the human ATC cell lines, we first performed immunofluorescence analysis on 8505C and TCO1 cells. Both cell lines expressed FGFR1, PDGFRβ, and VEGFR2 (Figure 1A). To examine the effects of lenvatinib on cell viability, 8505C and TCO1 cells were treated with various concentrations of lenvatinib and then subjected to the XTT assay. As shown in Figure 1B, treating cells with 0.1 to 50 μM lenvatinib for 48 h reduced cell viability in all human ATC cells in a dose-dependent manner.

The half-maximal inhibitory concentrations (IC_50_) for lenvatinib treatment of 8505C and TCO1 cells were 24.26 and 26.32 μM, respectively. Because lenvatinib is known to suppress receptor tyrosine kinases (RTKs) such as FGFR1, PDGFRβ, and VEGFR2, we performed Western blotting to investigate AKT and mitogen-activated protein kinase (MAPK) signaling downstream of the RTK signals (Figure 1C–E). Interestingly, phosphorylation of AKT was attenuated in a dose-dependent manner (Figure 1C,D), whereas ERK phosphorylation was amplified by E7080 treatment in both human ATC cell lines (Figure 1C,E). Additionally, we confirmed that the cleaved caspase 3 levels increased following the administration of lenvatinib (Figure 1F,G). These data indicate that the two human ATC cell lines express the E7080 target molecules, FGFR1, PDGFRβ, and VEGFR2. Furthermore, lenvatinib induces apoptosis by suppressing AKT signaling in human ATC cells.

### 3.2. Synergistic Effects of Lenvatinib (E7080) and an MEK Inhibitor (U0126) against ATC In Vitro

Because lenvatinib suppressed AKT signaling but did not inhibit MAPK signaling, we then investigated the effects of the combination of lenvatinib with U0126, an MEK inhibitor, in vitro. Lenvatinib alone, U0126 alone, and lenvatinib plus U0126 inhibited the proliferation of 8505C cells by 32%, 44%, and 66%, respectively, relative to the DMSO control (Figure 2A). Lenvatinib alone, U0126 alone, and lenvatinib plus U0126 inhibited the proliferation of TCO1 cells by 58%, 58%, and 72%, respectively, relative to the control (Figure 2B). Notably, potent synergistic effects of lenvatinib plus U0126 were observed in both cell lines. As shown in Figure 2C, when cells were treated with lenvatinib (5 μM) plus U0126 (5 μM) for 2 days, cell proliferation was inhibited in the two human ATC cell lines. To elucidate the molecular mechanism underlying the synergistic effects of lenvatinib and U0126, Western blot analysis was performed for AKT, ERK, and cyclin D1 (Figure 2D). The data revealed that lenvatinib inhibited AKT phosphorylation, while U0126 inhibited ERK phosphorylation and suppressed cyclin D1 expression (Figure 2E–J). These results indicate that lenvatinib and an MEK inhibitor exhibit synergistic inhibition of AKT and MAPK signaling in vitro.

### 3.3. Synergistic Effects of Lenvatinib (E7080) and an MEK Inhibitor (Selumetinib; AZD6244) against ATC in a Mouse Xenograft Model

Lenvatinib plus the MEK inhibitor U0126 showed synergistic inhibition of AKT and MAPK signals in vitro. To further examine the synergistic effects of lenvatinib with an MEK inhibitor in vivo, we used an 8505C-bearing ATC mouse model. Monotherapy with either lenvatinib (30 mg/kg/day) or selumetinib (30 mg/kg/day), an MEK inhibitor, inhibited tumor growth relative to the vehicle control (Figure 3A–C). Moreover, the combination of lenvatinib (30 mg/kg/day) and selumetinib (30 mg/kg/day) showed significantly greater antitumor activity than either monotherapy in terms of tumor volume (mm^3^), relative tumor volume from treatment day, and tumor weight (Figure 3A–D). However, pathological analysis showed that there were no morphological differences between the treatments (Figure 3E(a–d)). To confirm whether the combination of lenvatinib and selumetinib affected cell proliferation and apoptosis in our xenograft model, we performed immunohistochemistry analyses for Ki67 and cleaved caspase 3 (Figure 3E(e–l),F,G). Lenvatinib and selumetinib monotherapy significantly reduced Ki67 expression and increased cleaved caspase 3 levels (Figure 3E(f,g,j,k),F,G). Furthermore, the combination of lenvatinib (30 mg/kg/day) with selumetinib (30 mg/kg/day) treatment further reduced Ki67 expression significantly and increased cleaved caspase 3 levels (Figure 3E(h,l),F,G). Taken together, the combination of lenvatinib with selumetinib treatment had synergistic effects against tumor proliferation with apoptosis, which was based on AKT and ERK signal transduction pathways.

### 3.4. The Combination of Lenvatinib (E7080) with Selumetinib (AZD6244) Does Not Aggravate Side Effects in a Mouse Model

We also performed additional toxicity studies in the same mouse model. The combination of lenvatinib with selumetinib treatment did not cause greater body weight loss than monotherapy with either lenvatinib or selumetinib in addition to vehicle treatment (Figure 4A). However, lenvatinib monotherapy resulted in body weight loss compared to vehicle treatment. No toxicity was observed based on blood biochemical parameters such as TP, ALB, LDH, AST, ALT, T-BIL, BUN, CRE, AMY, Na, K, and Cl for combinatorial treatment compared to lenvatinib monotherapy, selumetinib monotherapy, and vehicle treatment (Figure 4B). Pathological analysis by HE sections showed that there were no morphological differences in the heart, lung, liver, kidney, and spleen of mice from the different treatment groups (Figure 4C). These results suggest that the combination of lenvatinib with selumetinib did not cause further toxicity in the mouse model.

## 4. Discussion

Although lenvatinib improved PFS and response rates in TC patients with RAI resistance, some patients could not continue treatment, and several patients required dose reduction or interruption [12]. Based on this study, the patients in the lenvatinib group required dose interruption (82.4% for lenvatinib vs. 18.3% for placebo) or reduction (67.8% for lenvatinib vs. 4.6% for placebo) owing to disease progression or toxicity [12]. The most common adverse effects developed during treatment—which led to dose interruption or reduction for patients receiving lenvatinib treatment with all grades of cancer or in patients with grade 3 cancer or higher—were hypertension (67.8% and 41.8%, respectively), diarrhea (59.4% and 8.0%), fatigue (59.0% and 9.2%), appetite loss (50.2% and 5.4%), and proteinuria (31.0% and 10.0%) [12]. Additionally, the results of subgroup analysis show that Japanese patients are more likely to require dose reduction (Japanese, 90%; overall, 67.8%) [21].

To understand the dose-dependent effects of lenvatinib, we investigated the response of RAI-refractory TC to lenvatinib in vitro using human ATC cells. Lenvatinib attenuated AKT signaling in human ATC cells in a dose-dependent manner. Meanwhile, lenvatinib did not suppress but, instead, activated MAPK signaling. In contrast, a previous study had shown that lenvatinib suppresses AKT and MAPK signaling in TC and lung cancer [22,23,24]. Ferrari et al. tested 8303C human ATC cells that bore p53, BRAF, PIK3CA, and TERT mutations, whereas Tohyama et al. used RO82-W-1 human differentiated thyroid cancer cells [22,23]. The differences in the cell lines, especially the differences in oncogenic mutations, may explain the divergent results we obtained in our study.

A recent study has revealed the importance of combination therapy for lenvatinib. Lenvatinib plus immunotherapy has been used for advanced solid tumors, such as adrenal cortical carcinoma, non-small-cell lung cancer, hepatocellular carcinoma, and head and neck squamous cell carcinoma [25,26,27,28]. Antiangiogenic therapy is known to reprogram the immunosuppressive tumor microenvironment and enhance immunotherapy [29,30]. To date, there is only one retrospective analysis of lenvatinib plus immunotherapy for TC, which achieved a 42% partial response rate in 12 patients with ATC [31]. Synergistic effects of immunotherapy with other agents (vinorelbine, doxorubicin, and paclitaxel) have been confirmed in preclinical models [31,32,33,34]. In the present study, we first demonstrated the potency of the combination of lenvatinib with an MEK inhibitor. The combination of lenvatinib and an MEK inhibitor showed synergistic effects in in vitro and in vivo experiments. A similar observation has been reported for sorafenib, which is known to be an inhibitor of Raf, RTK, VEGFR, and ERK signaling pathways for the therapy of various cancers [35]. Sorafenib treatment also upregulated MAPK signaling in these cancers, which suggests a Raf/ERK feedback loop. Thus, the combination of sorafenib with an MEK inhibitor has been considered and has shown potential in the treatment of gastric cancer, hepatocellular carcinoma, and renal cell carcinoma [36,37,38]. Interestingly, lenvatinib suppressed AKT signaling and induced apoptosis, whereas the MEK inhibitor suppressed MAPK signaling and inhibited cell proliferation in vitro. However, both lenvatinib and MEK inhibitor attenuated cell proliferation, as shown by decreased Ki67 expression (Figure 3E,F), and promoted apoptosis, as shown by increased cleaved caspase 3 expression (Figure 3E,G), in in vivo ATC models. The inconsistency between our results from in vitro and in vivo experiments may be caused by additional unknown antitumor mechanisms such as the alteration of tumor microenvironment in vivo.

Combination therapy may induce significant adverse reactions. Thus, we evaluated body weight, blood biochemical parameters, and histopathology in TC xenograft models for toxicity, but no significant differences were found in the combinatorial treatment group compared to single treatment or to the control in our preclinical model. Taken together, these results indicate that treatment with a combination of lenvatinib and an MEK inhibitor displays enhanced antitumor activity without additional toxicity. However, this study is a preclinical study. The dose-dependent antitumor and cytotoxicity effects are only assessed in mice. The synergistic effects by combinate use of chemical medicines are also only exhibited in mouse models. Thus, further investigations are needed for the clinical application of lenvatinib plus MEK inhibitors against ATC.

## 5. Conclusions

In this study, we tested the combination of lenvatinib with MEK inhibitors for ATC treatment in vitro and in vivo. We found that the combination enhanced the antitumor effect based on reduced tumor proliferation and increased apoptosis, which is via AKT and ERK signal pathways. Furthermore, we found that the combination did not have significant adverse effects. Further investigation should be done before clinical application.

## Figures and Tables

**Figure 1 cancers-13-00862-f001:**
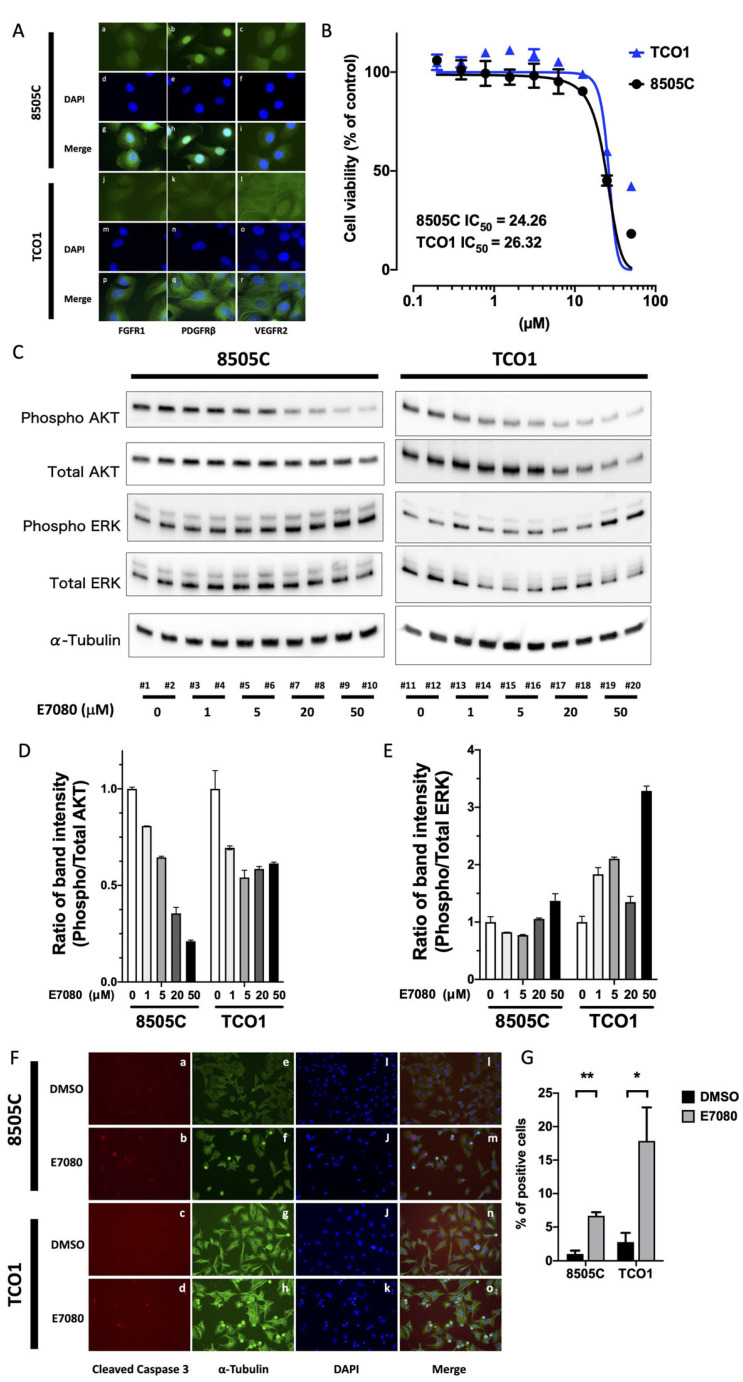
The antitumor effect of lenvatinib (E7080) monotherapy in an in vitro model. (**A**) Immunofluorescence analysis of 8505C and TCO1 human ATC cells confirmed the expression of FGFR1 (panels **a,j** green), PDGFRβ (panels **b,k** green), and VEGFR2 (panels **c,l** green). DAPI (panels **d–f, m–o** blue) was used for nuclear staining. They were merged and are shown in panels **g**–**i, p**–**r.** (**B**) Effects of different concentrations of lenvatinib on cell viability. (**C**) Western blot showing phosphorylation of AKT and ERK in 8505C or TCO1 cells with 1 to 50 μM E7080 for 24 h or DMSO as control. (**D**,**E**) The band density was quantified and compared among different doses. (**F**) Cleaved caspase 3 level was confirmed by immunofluorescence assay on 8505C and TCO1 cells treated with 50 μM lenvatinib for 24 h or with DMSO as control (panels **a**–**d**). α-Tubulin and DAPI are shown in green (panels **e–h**) and blue (panels **i**–**k**), respectively. They were merged and are shown in panels **l**–**o**, indicating the presence of cleaved caspase 3 in the nuclei of 8505C and TCO1 cells. (**G**) Quantification of cleaved caspase 3-positive cells. 8505 and TCO1 cells treated with 50 μM lenvatinib for 24 h had significantly higher counts of cleaved caspase 3-positive cells than cells treated with DMSO. * *p* < 0.05, ** *p* < 0.01.

**Figure 2 cancers-13-00862-f002:**
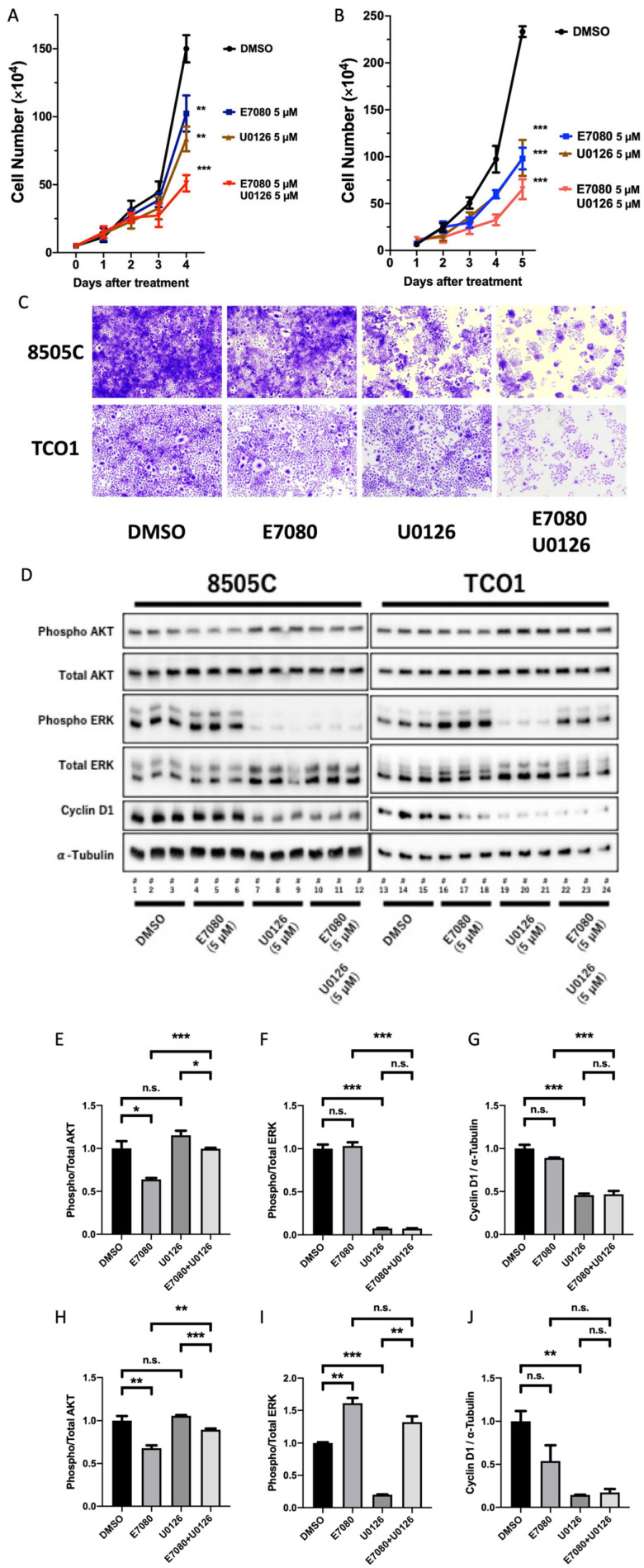
The synergistic effect of lenvatinib (E7080) plus MEK inhibitor in vitro. (**A**–**C**) Lenvatinib plus an MEK inhibitor, U0126, inhibited cell proliferation more potently than monotherapy with either lenvatinib or U0126 in 8505C and TCO1 cell lines (** *p* < 0.01, *** *p* < 0.001). (**D**) Western blot analysis shows that lenvatinib inhibited phosphorylation of AKT and U0126 inhibited phosphorylation of ERK and cyclin D1 in 8505C and TCO1 human ATC cells. Quantification of band density is shown for 8505C (**E**–**G**) and TCO1 cells (**H**–**J**). * *p* < 0.05, ** *p* < 0.01, *** *p* < 0.001, n.s.: not specified.

**Figure 3 cancers-13-00862-f003:**
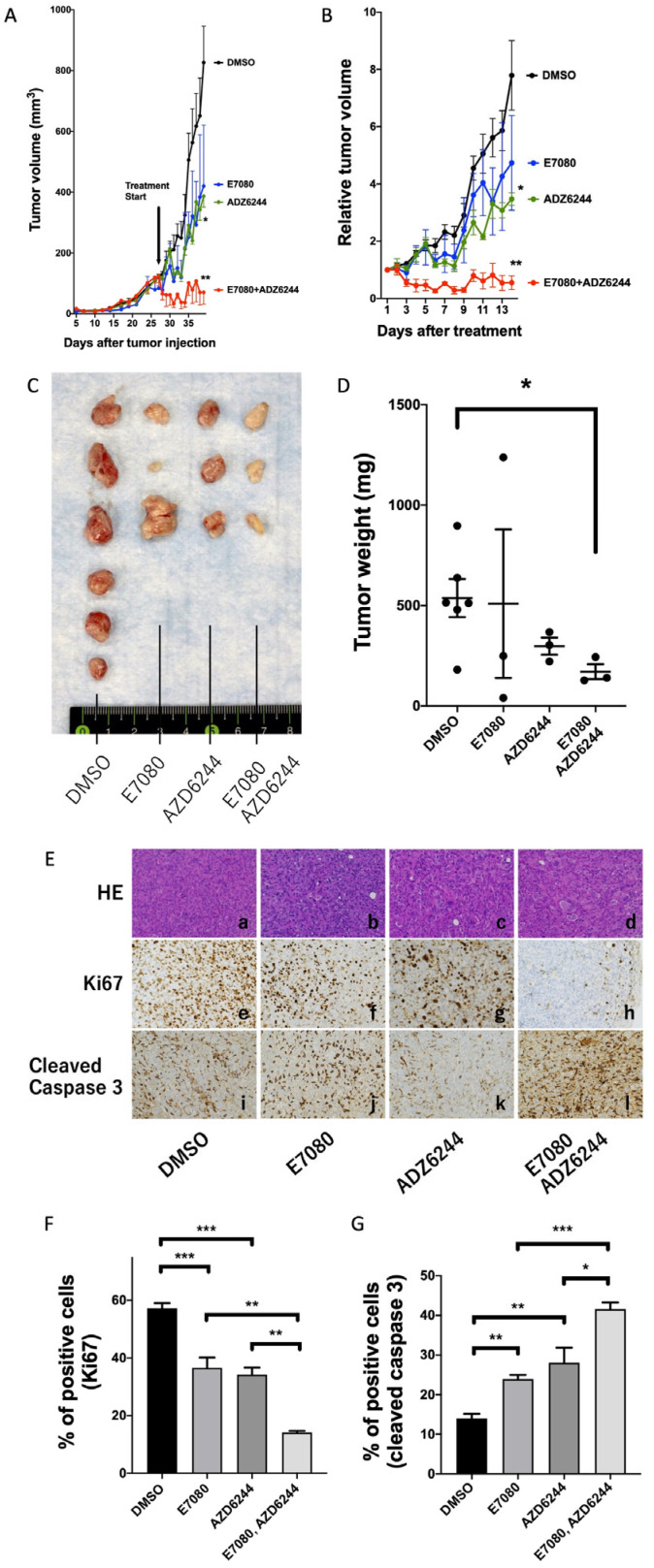
Antitumor effects of lenvatinib and selumetinib in 8505C-inoculated athymic BALB/c nude mice. An equal number of 8505C cells (1 × 10^6^ cells) was injected into the flanks of each mouse before treatment. When tumors started to develop (average tumor size reached 100 mm^3^), lenvatinib plus selumetinib, lenvatinib alone, selumetinib alone, or the vehicle was orally administered to the mice for 14 days (30 mg/kg/d). (**A**) Tumor volume, (**B**) tumor growth rate, (**C**) tumor size, and (**D**) tumor weight were decreased by treatments. (**E**) Representative images of H&E-stained tumor sections (panels **a**–**d**) are shown. There were no marked morphological differences among the tumors from mice treated with the vehicle (panel **a**), with lenvatinib alone (panel **b**), with selumetinib alone (panel **c**), and with the combination of lenvatinib and selumetinib (panel **d**). Lenvatinib monotherapy reduced Ki67 expression (panel **f**) and increased cleaved caspase 3 levels (panel **j**). Selumetinib reduced both Ki67 and cleaved caspase 3 levels (panels **g**,**k**). Furthermore, the combination of lenvatinib with selumetinib treatment strongly reduced Ki67 expression and increased the levels of cleaved caspase 3 (panels **h**,**l**). Quantification of Ki67 (**F**) and cleaved caspase 3-positive cells (**G**) is shown. * *p* < 0.05, ** *p* < 0.01, *** *p* < 0.001.

**Figure 4 cancers-13-00862-f004:**
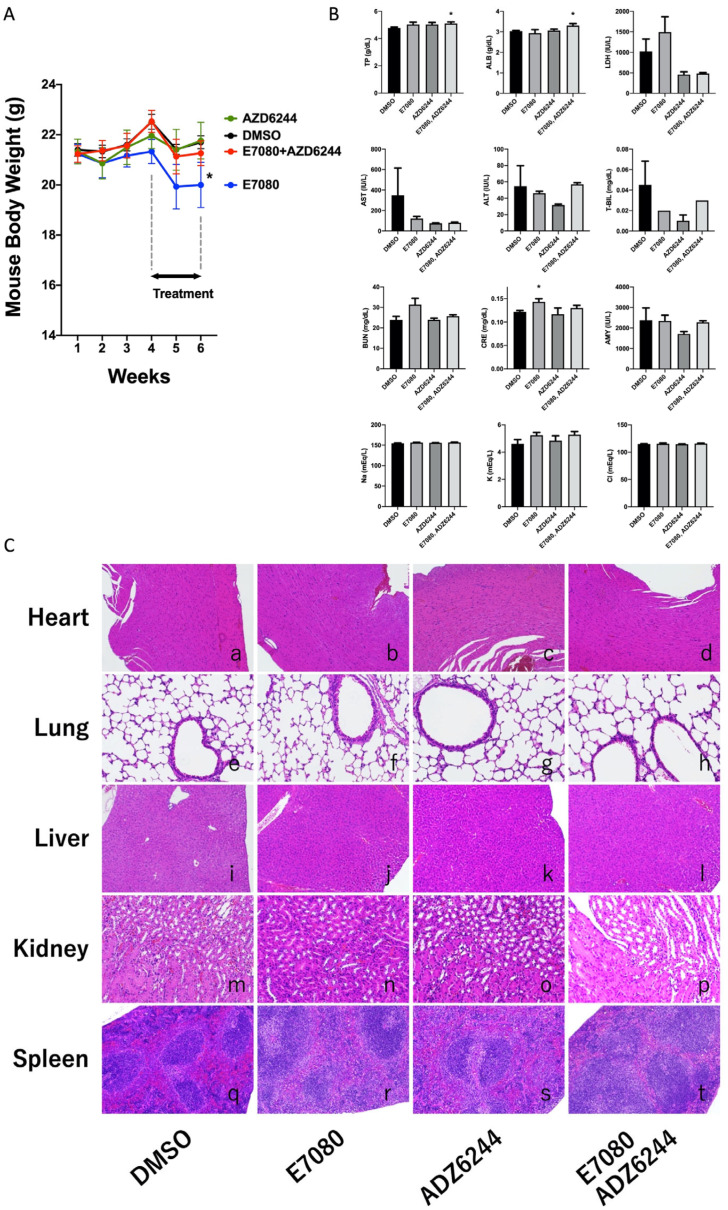
Toxicity assessment in 8505C-inoculated athymic BALB/c nude mice. (**A**) Lenvatinib plus selumetinib did not cause greater body weight loss than monotherapy with either lenvatinib or selumetinib or vehicle treatment, but lenvatinib monotherapy resulted in body weight loss compared to vehicle treatment (* *p* < 0.05). (**B**) No toxicity was observed in blood examination, TP, ALB, LDH, AST, ALT, T-BIL, BUN, CRE, AMY, Na, K, and Cl levels in all treatments (* *p* < 0.05). (**C**) Pathological analysis (H&E sections) showed there were no morphological differences in major organs obtained from mice with different treatments.

## Data Availability

The data presented in this study are available in this article and Appendix A.

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
