# Peer review of "Synergistic Effects of Lenvatinib (E7080) and MEK Inhibitors against Anaplastic Thyroid Cancer in Preclinical Models"

_cancers, 2021, doi:10.3390/cancers13040862_

Round 1
Reviewer 1 Report
This is a detailed report of pre-clinical mechanistic effects of lenvatinib in combination with MEK inhibitors on growth, apoptosis, and kinase activity in thyroid cancer cell cultures, animal xenograft models and human subjects. The results show significantly improved synergy over monotherapy alone. Overall, the data are compelling for combined therapy. There is one area of confusion for me which is on page 9 beginning with line 292:
Lenvatinib monotherapy reduced Ki67 expression and increased 291
cleaved caspase 3 levels (Figure 4Ef,j). In contrast, selumetinib reduced Ki67 and cleaved 292
caspase 3 levels, indicating that selumetinib did not contribute to apoptosis (Figure 4Eg,k). 293
Furthermore, the combination of lenvatinib (30 mg/kg/day) and selumetinib (30 294
mg/kg/day) significantly reduced Ki67 expression and increased cleaved caspase 3 levels 295
(Figure 4Eh,l). Taken together, the combination of lenvatinib and selumetinib had syner- 296
gistic effects against tumor proliferation and on apoptosis, and these effects are dependent 297
on AKT and ERK signaling
It is not clear to me how the authors conclude that Sulemetinib did not contribute to apoptosis since Ki67 was reduced and cleaved caspase was increased. Please clarify.
Author Response
Thank you very much for giving us precious comments. To clarify the vivo mechanism of Sulemetinib, we have performed further analysis of IHC data. We added counting data in new figure 3F and 3G, and revised discussion part on page 17 beginning with line 361.
Reviewer 2 Report
The manuscript titled „Synergistic Effects of Lenvatinib (E7080) and MEK Inhibitors against Radioiodine-Refractory Thyroid Cancer in Preclinical Models” has two, partly related sections: a retrospective analysis of efficacy and tolerability of lenvatinib in 25 patients suffering from radioiodine-refractory thyroid cancer (among them 7 had anaplastic cancer) and an elaborated preclinical experimental work investigating the effect of a combined treatment with lenvatinib and MEK inhibitors in anaplastic thyroid cancer (ATC) cell lines and in ATC mouse model.
The possible reason to include the retrospective, real-life clinical study is to confirm the need for further treatment options. However, it is out of question. Moreover, the clinical part is not well detailed, many questions are raised, for example why the very low-dose, 4 mg of lenvatinib was regularly used, how the adverse events were managed by concomitant medication, etc. I suggest to consider the omission of this clinical part, because it is not an integrated part of the manuscript.
The second part of the paper investigated a very important question: what may be a synergistic combination of targeted therapies in ATC, which is a lethal disease. We have promising preliminary results of phase II clinical trials investigating the effect of lenvatinib in this patient group but it is obvious that this treatment is not curative. Ongoing clinical trials investigate the therapeutic effect of lenvatinib and pembrolizumab (https://www.clinicaltrials.gov/ct2/results?cond=anaplastic+thyroid&term=lenvatinib&cntry=&state=&city=&dist=), so to find more targets is desirable.
The authors found that lenvatinib suppressed AKT signaling but did not inhibit MAPK signaling in ATC cell lines, so a combination of lenvatinib with a MEK inhibitor would be an effective treatment. It was convincingly proved in vitro, in ATC cell lines and in vivo, in a nude mouse model. It is an important result and the experiments are well presented, however, the concept is not new, the synergistic effect of sorafenib and selumetinib was previously published in preclinical models of gastric cancer, hepatocellular carcinoma and renal cell cancer.
The title of the manuscript is too general, because the study was done on ATC cell lines, I suggest to change the title: „Synergistic Effects of Lenvatinib (E7080) and MEK Inhibitors against Anaplastic Thyroid Cancer in Preclinical Models”
My other critical comments:
page 1, row 22: “lenvatinib, is an oral multitargeted tyrosine kinase that” – the “inhibitor” is missing
page 1, row 41: „with reported incidences rates of 12.9% in the USA (2011 to 2015) and 25.8 per 100,000 individuals in Puerto 41 Rico (2011 to 2015) [1].” – it is “12.9 per 100,000 individuals”
page 6, Figure 2. E part: The legend of Y axis is not correct; it should be ERK instead of AKT
and few other typing errors…
Author Response
The manuscript titled „Synergistic Effects of Lenvatinib (E7080) and MEK Inhibitors against Radioiodine-Refractory Thyroid Cancer in Preclinical Models” has two, partly related sections: a retrospective analysis of efficacy and tolerability of lenvatinib in 25 patients suffering from radioiodine-refractory thyroid cancer (among them 7 had anaplastic cancer) and an elaborated preclinical experimental work investigating the effect of a combined treatment with lenvatinib and MEK inhibitors in anaplastic thyroid cancer (ATC) cell lines and in ATC mouse model.
The possible reason to include the retrospective, real-life clinical study is to confirm the need for further treatment options. However, it is out of question. Moreover, the clinical part is not well detailed, many questions are raised, for example why the very low-dose, 4 mg of lenvatinib was regularly used, how the adverse events were managed by concomitant medication, etc. I suggest to consider the omission of this clinical part, because it is not an integrated part of the manuscript.
Thank you for your careful reviewing to our manuscript, and we appreciate to reviewer’s very valuable suggestions. As the clinical part is not well detailed and did not support our conclusions of experiment part, we have decided to exclude clinical part according to the reviewer's comments. We could focus on combination therapy E7080, lenvatinib with MEK inhibitor.
As the clinical part was totally omitted, one of authors, Sachiko Hayata who provide clinical data, was moved from authors to acknowledgement for her valuable discussion of lenvatinib in clinical patients.
The second part of the paper investigated a very important question: what may be a synergistic combination of targeted therapies in ATC, which is a lethal disease. We have promising preliminary results of phase II clinical trials investigating the effect of lenvatinib in this patient group but it is obvious that this treatment is not curative. Ongoing clinical trials investigate the therapeutic effect of lenvatinib and pembrolizumab (https://www.clinicaltrials.gov/ct2/results?cond=anaplastic+thyroid&term=lenvatinib&cntry=&state=&city=&dist=), so to find more targets is desirable.
The authors found that lenvatinib suppressed AKT signaling but did not inhibit MAPK signaling in ATC cell lines, so a combination of lenvatinib with a MEK inhibitor would be an effective treatment. It was convincingly proved in vitro, in ATC cell lines and in vivo, in a nude mouse model. It is an important result and the experiments are well presented, however, the concept is not new, the synergistic effect of sorafenib and selumetinib was previously published in preclinical models of gastric cancer, hepatocellular carcinoma and renal cell cancer.
Absolutely, the concept may not be new. However, we believe that this study includes important novel discoveries as following reasons. At first, we first showed what is a candidate of novel targeted therapies in ATC treatment. Secondly, we also first reported what is a synergistic combination plus lenvatinib, E7080, in cancer therapies. Therefore, this study provides many useful information for readers not only endocrinologist including thyroid surgeon but also molecular biologist and clinical oncologist treated with gastric cancer, hepatocellular carcinoma and renal cell cancer. We have revived discussion and conclusions.
The title of the manuscript is too general, because the study was done on ATC cell lines, I suggest to change the title: „Synergistic Effects of Lenvatinib (E7080) and MEK Inhibitors against Anaplastic Thyroid Cancer in Preclinical Models”
To express this study, I have changed the title to “Synergistic Effects of Lenvatinib (E7080) and MEK Inhibitors against Anaplastic Thyroid Cancer in Preclinical Models”.
My other critical comments:
page 1, row 22: “lenvatinib, is an oral multitargeted tyrosine kinase that” – the “inhibitor” is missing
Thank you for point out. I have corrected typos.
page 1, row 41: „with reported incidences rates of 12.9% in the USA (2011 to 2015) and 25.8 per 100,000 individuals in Puerto 41 Rico (2011 to 2015) [1].” – it is “12.9 per 100,000 individuals”
I have revised to “12.9 per 100,000 individuals”.
page 6, Figure 2. E part: The legend of Y axis is not correct; it should be ERK instead of AKT
and few other typing errors…
Thank you for point out. We have re-read and corrected typos.

Round 2
Reviewer 2 Report
I accept the authors' responses without any further comment.